# A congruence perspective on how human and social capital affect learning capability and innovation

Rui Sun[1☯], Shuwen Li [2☯]*, Wei Liu[3]

1 School of human resource, Chinese Academy of Personnel Science, Beijing, China, 2 School of Economics and Management, Tongji University, Shanghai, China, 3 Central University of Finance and Economics, Beijing, China

☯ These authors contributed equally to this work.
* lishuwen7730@163.com

## Abstract

Human capital and social capital are vital for sustainable development of organization, but existing studies are inadequate to explore the synergistic effect of them on organizational behaviors or organizational outcomes. The research employed multiple-source question-naire to collect data of more than 400 R&D firms from leaders and corresponding employees in Chinese context. Bootstrapping method and response surface were used to analyze the associations between all the parameters. The results showed that: (1) in the case of the in-congruence between human capital and social capital, learning capability of R&D firms is at highest level when human capital is at low level and social capital is at high level; (2) learning capability partially mediates the impact of human capital and social capital on innovation performance; (3) environmental dynamism and environmental competitiveness have a joint moderating effect on the relationship between learning capability and innovation performance.

## Introduction

Nowadays, organizations are seeking to develop in greatly changing domestic and international environment, and the uncertainty of the environment has become the new normal in organizational development [1–3]. In such a situation, the principal challenge faced by firms, especially high-tech firms, is how to navigate the uncertainty of the environment [4]. There has been considerable evidence demonstrating that innovation is the key to tackle uncertainty and the important way to gain sustainable development of organization [5, 6]. Thus, how to stimulate effective and sustainable innovation has been a significant breakthrough in dynamic and competitive environment. Recently, exploring the antecedents of innovation has become one of the research hotpots in the field of management [7, 8], and a large number of studies indicated that human capital and social capital, broadly defined as organizational resources to provide knowledge, skills and social network structures [9, 10], play an important role in stimulating innovation and helping organizations to meet environmental challenges. However,

**Data Availability Statement:** All relevant data are within the paper and its Supporting Information files.

**Funding:** Major projects of China Social Science Foundation (17ZDA057)-Rui Sun.

**Competing interests:** The authors have declared that no competing interests exist.

though the main effects of these two types of capital on innovation respectively have been widely confirmed by prior researches [11, 12], less far is known about how the synergistic effect of human and social capital on innovation or other organizational outcomes. In fact, human capital and social capital are no independent existence each other in management practices, and they are coexistence and influenced each other.

According to resource-based view, human capital and social capital are critical organizational resources that are inimitable, rare, valuable and non-substitutable [13–15], and they can be converted into other forms of resources [16] through organizational engagement and motivational antecedents [17]. However, some inconsistencies, how these two types of capital interact in shaping organizational relevant outcomes, have been revealed in previous few findings. There are three contradicting theoretical claims on the relationship between human capital and social capital. Some studies claim that the relationship is competitive, as network resource is more vital than knowledge resource for organizations that are in short of human capital [18, 19]. But some researchers posit human capital and social capital are complementary each other, as knowledge and skills can be beneficial for strengthening the social network, and vice versa [19–21]. Besides, others suggest the relationship is nether competitive nor complementary, as the relationship is contingent on the organizational resource support provided by network structure [22, 23]. Thus, existing studies have not got a consistent conclusion on the relationship between human capital and social capital, and the effect of capital on organizational outcomes, especially lacking of the empirical analysis under different circumstances.

In R&D firms, innovation was regarded as one of the most important contributors to the organizational development and survival [24]. Various features in R&D firms, such as innovational route fuzzy, innovational time pressing and innovational situation changing, are presented in the process of creative activities, which determine whether the organization is more sensitive to changes in the environment. These characteristics have a direct impact on the organizational capability, especially organizational learning capability, and all the resources (e.g., knowledge) around the organization are conducted by R & D enterprises to increase strategic certainty [25, 26], plan accuracy and skill exploration [27], and decrease resource threat of internal and external environmental changes in organizational innovation and creation activities.

In this study, we aim to develop a theoretical model, where the influence procedure and boundary condition of human capital and social capital on innovation performance are examined simultaneously in R&D firms. As we explain in more details below, the current study contributes to the existing understanding about R & D enterprises in three main streams: Firstly, we focus on the joint effect of human capital and social capital on learning capability, bring insight into the relationship between two types of capital being competitive or complementary, and extend the research on the association between capital and organizational learning. Secondly, we highlight the mediating role of learning capability between capital and innovation. Thirdly, we shed light on the mechanisms of organizational internal and external environment in the process of resource transformation.

## Research model and hypotheses

We develop a mediated moderation model suggesting the relationship between all the parameters. Fig 1 presents research model.

### Human capital, social capital and learning capability

Capital, especially human capital and social capital, has always been viewed as one of important factors determining organizational behaviors and organizational outcomes, because these

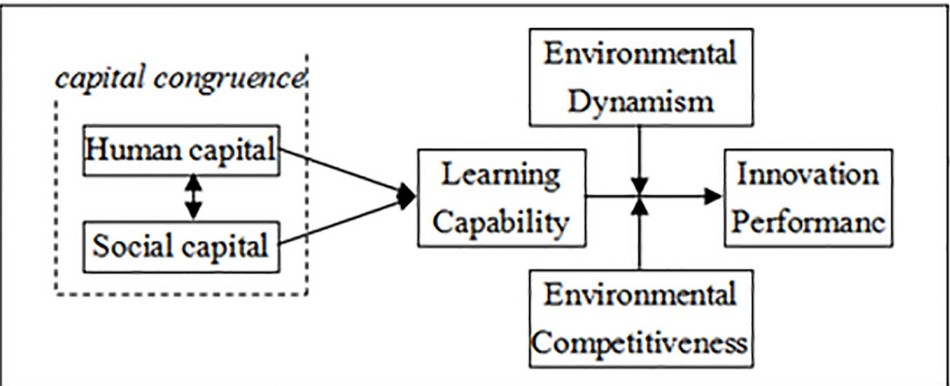

**Fig 1. Hypothesized research model.**

two kinds of capital are prominently featured as organizational resource. Human capital, defined as the resource related to expertise and knowledge [28], represents knowledge resources converted easily across the organizational settings [29, 30]. Social capital, defined as the resource related to network ties [28], represents social network resources contributing to develop new business and meet new challenge in response to environmental changes [18, 23, 31]. According to resource-based view, human capital and social capital are heterogeneous redundant resource with potential competitive advantage in the organization, and heterogeneity of resources determines the difference in transformation of organizational resources [13–15].

The organization itself does not possess the ability to learn, but it can acquire learning capability depending on the members of the organization to share knowledge with each other and to learn from other organizations, customers, research institutes, and suppliers [32]. That is, the acquisition of organizational learning resources comes from the multiple paths between the organization and the outside. Chipika and Wilson [33] also highlighted the importance of networking for organizational learning capability. Alegre & Chiva [34] have conceptualized learning capability as the skills that facilitate the usage and dissemination of organizational knowledge.

Based on competitive advantage theory, organization is an aggregate composed of a series of unique resources, and the nature of organizational development is the process of integrating and reconfiguring unique resources [35]. Teece [26] further figured out the foundation of the enterprise's internal ability comes from the organizational core assets (e.g. human and social capital). Studies on organizational learning have also emphasized that the function of organizational learning capability mainly depends on the exchange and integration of existing information, knowledge, and various ideas [36]. Moreover, R & D firm is a typical organization with high level of resource demand, especially knowledge resource, and knowledge activities of particular importance in R&D firms including the sharing of information and know-how among a wide range of individuals, and the activities contribute to increasing knowledge resources acquisition and absorptive capacity [37]. Hansen [38] posits that strong ties have more significant and positive effect on sharing of knowledge, whereas weak ties are not beneficial because of non-redundant contacts. Since social capital stems from norms for collaboration, interaction and the sharing of ideas [16, 39], and tends to promote the frequency of knowledge resources interaction and absorptive capacity within organization [40]. Drawing on resource-based view and the related literature [30], we contend that human capital and social capital are heterogeneous organizational resources, and learning capability is the skill utilizing resources.

Only when heterogeneous resources are injected into the organizational system can organizations strengthen the ability to learn and apply knowledge resources. Therefore, it is hypothesized that:

> **Hypothesis1**: *(a) Human capital and (b) social capital will be positively related to learning capability.*

## Capital congruence and learning capability

Several studies have viewed intellectual capital as organizational knowledge resources, and consider it as the sum of all the knowledge in organization [41]. Subramaniam and Youndt [7] argued that human capital and social capital were two most vital aspects of intellectual capital, and there was a significant synergy and matching relationship between them [42]. In other words, human capital may provide diverse thoughts and ideas for firms to develop social capital, the potential of social capital connects these thoughts to make unusual and unforeseen combinations and enhances the returns of human capital [43]. However, there are various paring situations (e.g. congruence or incongruence) for the matching relationship between human and social capital. Thus, we divide the level of human capital (HC) and social capital (SC) into four capital paring situations (in Fig 2): ① Low HC-Low SC; ② Low HC-High SC; ③ High HC-Low SC; ④High HC-High SC. Among these paring situations, ① indicates low level of capital congruence, ④ indicates high level of capital congruence, ② ③ indicate capital incongruence. The capital paring situations reflected different synergy and matching ability of two types of capital. In this study, ① ④ are defined as complementary between human and social capital, and ② ③ are defined as competitive between them.

In high HC-high SC situation, the high level of balance is obtained in the process of transformation of capital resources. Based on conservation of resource theory, organizations and employees with more capital resources are more likely to increase resource-input, promote the process of resource-transformation, and then translate capital resources into knowledge resources [44]. In low HC-low SC situation, although human capital and social capital have also been balanced, they cannot meet the resource requirements in the process of resource transformation at a low level. As an important way to acquire knowledge resources, low level of capital congruence can make employees keep their resources, reduce their communication and information sharing, hinder the search and interaction of external information, and further reduce organizational application ability of knowledge resources. Therefore, it is hypothesized that:

> **Hypothesis2**: *The interaction between human capital and social capital predicts learning capability, such that learning capability is highest when both human capital and social capital are high.*

## The mediation effect of learning capability

Drawing on Sophie and Lus's (2010) theoretical development, Akhavan and Hosseini [11] propose that the capital can influence innovation of R&D firms through shaping organizational capability. Snell and Dean [45] argue the hallmarks of human capital are skilled, bright and creative employees with expertise, who constitute the predominant resource for new knowledge and ideas. Tushman and Anderson [46] also highlight the employees with abundant skills and knowledge are more likely to change organizational routines and question prevailing norms, then to shorten production cycle, update organizational process and alter management methods. Hill and Rothaermel [47] suggest it is much important for knowledge

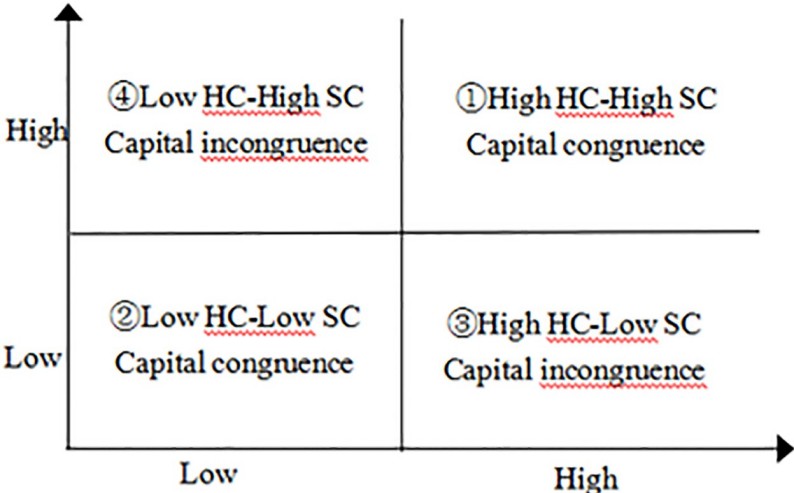

**Fig 2. Capital matching situation.**

employees to expose an organization to technology boundaries which enhance the capability to deploy and absorb knowledge, then to advance the transformation of prevailing knowledge and the application of resources. Social capital is the connection that combines new ideas one another, as Hargadon and Sutton [48] describe the process of combing as "brokering", and argue the unconventional combining is a key to gain innovational resources. Furthermore, to develop new products, processes and administration that contribute to the organizational innovation, R&D firms must capture, interpret, and deploy knowledge resources [49]. Based on existing literatures, we contend learning capability as an organizational ability which deploys and absorbs knowledge, trappings of human capital is knowledge (e.g. new ideas), the core of social capital is combining, connecting and dissemination different knowledge or thoughts, and the hallmark of innovation is an outcome applied knowledge. In other words, in the process of resource transformation, human and social capital as different knowledge resource input, innovation performance as knowledge resource output, and learning capability is a convert hub or re-shape impetus of knowledge resource. Therefore, it is hypothesized that:

> **Hypothesis3**: *(a) Human capital and (b) social capital will be positively related to innovation performance.*

> **Hypothesis4**: *Learning capability will mediate the relationship between (a) human capital, (b) social capital and innovation performance.*

## The joint moderation effect of internal and external environment

In the practices of organizational management, organizations often face two kinds of external environmental characteristics with opposite tensions or paradoxes, and some scholars argue they are environmental dynamism and environmental competitiveness [50]. Environmental dynamism refers to the rate of change and the degree of instability in the environment [4], defined as organizational internal environment; environmental competitiveness refers to the degree of competition reflected in the number of competitors and the number of areas with competition [4], defined as organizational external environment. Based on dynamic capability theory, as the two sides of an organizational environment, the ambidextrous environment

(environmental dynamism and competitiveness) is not naturally balanced in the process of transforming resources, but always in the dynamic synergy and matching condition [51].

Based on Liu, Chen and Yao [52], organizational environment is considered as the important contingency factor in the process of the organization and management system resources transition. While according to resource-based view and conservation of resource theory, human and social capital are concretely expressed as knowledge and network resource respectively; innovation performance is regarded as intellectual resources of innovation, as proposed by Barrick, Thurgood, Smith and Courtright [17] that different forms of resources can be converted with each other through organizational engagement and motivational antecedents. With the internal dynamics and external competition from low level to high level, environmental changes will optimize the organizational results and push forward the organizational resources [53]. In other words, on one hand, the high level ambidexterity environment makes some products, processes and administration obsolete and require new ones to be developed [54]. In this case, the surrounding resources (such as knowledge) will be mobilized to promote product and technology innovation through organizational learning and exploring new markets and technologies, to strengthen the knowledge absorptive capacity, to minimize the threat of resource, to increase strategic certainty, to plan accuracy and skill exploration [25–27], then to form gain-spiral of organizational resources and to meet organizational needs of intellectual resources [55]. On the other hand, organizations may foster a climate of satisfaction with status quo in a low uncertainty and competitive environment, and employees are inclined to preserve their resources. In this case, organizational innovation will be hampered as consistent values and learning atmospheres can not be built in organizations, then to form loss-spiral and reduce available resources for innovations [56]. Therefore, it is hypothesized that:

> **Hypothesis5**: *Environmental dynamism and environmental competitiveness will jointly positively moderate the relationship between learning capability and innovation performance, such that the effect, among capital congruence, learning capability and innovation performance, will be strongest positive when Environmental dynamism and environmental competitiveness are high.*

## Method

### Procedures and samples

We collected data from R&D employees and leaders from firms of the information technology industry in China, and these firms were among the most competitive ones in the industries of electronic technology, software development, electronic communications, new materials, and machinery manufacturing. Firms employing more than 25 employees were conducted to participate in this study as they were able to implement formal R&D and innovation system. We investigated equal proportion of both employees and leaders in order to minimize common method bias. Participation in the project was absolutely voluntary, all answers were completely confidential (with no obligation to respond) and the data were analyzed anonymously. R&D employees responded to a survey in which they were asked to evaluate human capital, social capital and learning capacity. R&D leaders responded to another survey with items used to assess environmental dynamism, environmental competitiveness and innovation performance. Respondents in the study were voluntary to participate the survey.

We delivered survey forms to 694 firms and 456 were given back. These firms were from various technology sectors, including electronic communication (N = 81 firms), machinery manufacturing (N = 101 firms), biopharmaceutical (N = 71 firms), chemical food (N = 146 firms), software development (N = 35 firms), and others (N = 22 firms). Among the firms,

16.7% were at the initial stage, 44.1% were at the developing stage, 36% were at the maturity stage, and 3.2% were at the decline stage; organizational age was as follows: 1–2 years (19.1%), 3–5 years (15.8%), 6–10 years (27.4%), more than 10 years (37.7%); for organizational size, 15.6% had 25–50 employees, 21.1% had 50–200 employees, 21.9% had 200–500 employees, 10.5% had 500–1000 employees, 30.9% had more than 1000 employees.

## Measurements

**Human capital and social capital.** We used five-item human capital measure from Subramaniam and Youndt [7] to asses each participants' perception of the knowledge, expertise and skill. Likewise, the five items measuring social capital drew from the scale of Subramaniam and Youndt [7], these items reflected the overall ability to share and leverage knowledge among employees. Participants were asked to assess on a 5-point scale (1 = never to 5 = always) with the instruction. A sample item is "our employees are skilled at collaborating with each other to diagnose and solve problems". The internal consistency reliability for these constructs were 0.908 and 0.893 respectively. The factor loading for human capital is between 0.716 and 0.771, and for social capital is between 0.583 and 0.783.

**Learning capability.** To measure the extent of learning capability, we used fourteen-item organizational learning capability scale from Alegre and Chiva [34], these items reflected interactions among and between actors (individuals and groups). The interactions among actors include participative decision making, dialogue and interaction with the external environment, the interactions between actors include risk taking and experiment. We used the average score of fourteen-item to measure learning capability. A sample item is "initiative often receives a favorable response here, so people feel encouraged to generate new ideas". The internal consistency reliability for this construct was 0.908. The factor loading for learning capability is between 0.636 and 0.915.

**Environmental dynamism and environmental competitiveness.** These variables were measured with three-item from Jansen, Bosch and Volberda [4], respectively. These items for dynamism respected tapped into the instability and the rate of change of organizational internal environment, and the items for competitiveness measured organizational external environment in the number of competitors and competitive areas. The internal consistency reliability for these constructs were 0.806 and 0.825 respectively. The factor loading for environmental dynamism is between 0.624 and 0.855, and for environmental competitiveness is between 0.629 and 0.835.

**Innovation performance.** We used seven-item measure from Jiménez-Jiménez and Sanz-Vallel [57] to assess the changes in the products, processes or administrative systems developed by the company. The two items for production innovation were to measure new product introduced and R&D expenditure in new products; the two items for process innovation were to measure changes in the process introduced and pioneer disposition to introduce new process; the three items for administrative innovation were to measure novelty of the management systems and search of new management systems by directives. We used the average score of seven items to measure innovation performance. Participates were asked to assess on a 5-point scale (1 = below competitors, 3 = similar competitors to 5 = above competitors) with the instruction. The internal consistency reliability for this construct was 0.829. The factor loading for innovation performance is between 0.591 and 0.783.

**Control variables.** According to precious studies [4, 7, 34], the organizational differences have been found to affect organization innovation and environment. Thus, we used organizational type, organizational age, organizational developing stage and organizational size as the control variables.

## Data analysis

Using Lisrel8.7, we conducted confirmatory factor analysis (*CFA*) of all variables to assess the validity. The hypothesized six-factor model test indicated standardized root mean square residual (*RMSEA*) = 0.056, comparative fit index (*CFI*) = 0.87, normed fit index (*NFI*) = 0.86, incremental fit index (*IFI*) = 0.87, items loading were significant (*p<0.01*), which verified the validity of measurement.

## Results

Descriptive statistics and correlations are presented in Table 1. Human capital and social capital had a positive correlation with learning capability (*r* = 0.602, *p<0.01*; *r* = 0.704, *p<0.01*), as well as a significantly positive correlation with innovation performance (*r* = 0.570, *p<0.01*; *r* = 0.442, *p<0.01*). Learning capability was positively related to innovation performance (*r* = 0.443, *p<0.01*). The correlations between variables were less than their corresponding square root of AVE.

Table 2 presents the results of hierarchical regression of main and mediation effects. The predictors were centered by subtracting their mean values before computing the interaction terms. As indicated by the R square, adjusted R square and F values, the main and mediation effects explained a proportion of the variance in hierarchical regression. Moreover, to test for multiple-collinearity, we examined variance inflation factors *(VIF)* in each regression equation. The maximum of VIF was 2.111, below rule of thumb cut off of 10. The results in model1 and model2 show that human capital and social capital were significantly associated with learning capability (*β = 0.593, p<0.01; β = 0.729, p<0.01*). Thus hypothesis1 was supported. Model3 shows that the interaction between human capital and social capital (capital congruence) had significantly positive association with learning capability (*β = 0.058, p<0.1*). Hypothesis2 was supported. Model4 and model5 show that human capital and social capital were significantly associated with innovation performance (*β = 0.563, p<0.01; β = 0.506, p<0.01*). Hypothesis3, human capital and social capital will be positively related to innovation performance, was supported. As shown in model6 and model7, the direct effects of human capital and social capital on innovation performance decreased after introducing learning capability into regression equation, and the association between learning capability and innovation performance was significant (*β = 0.152, p<0.01; β = 0.193, p<0.01*). Meanwhile, human capital and social capital had positively influence on innovation performance(*β = 0.472, p<0.01; β = 0.365, p<0.01*). Thus learning capability partly mediates the effects of human capital, social capital on innovation performance. Hypothesis4 was supported.

To further test the mediation effects, we used the bootstrapping method (95% confidence interval, sample = 5000) and Sobel test to examine the direct and indirect effects. The results were displayed in Table 3. Bootstrapping test showed both indirect effect (*β = 0.067,CI [0.029,0.106]; β = 0.154,CI[0.095,0.216]*) and direct effect (*β = 0.341,CI[0.274,0.408]; β = 0.218, CI[0.122,0.314]*) of human capital and social capital on innovation performance, and the 95% bias-corrected confidence interval excluded zero. The sobel test values of human capital and social capital were 3.201(*p<0.01*) and 4.381(*p<0.01*). Hypothesis4 was supported further.

To examine synergistic effect of human capital and social capital, we used the response surface in the study. Firstly, polynomial regression was conducted, and its equation was $Z = b_0 + b_1 X + b_2 Y + b_3 X^2 + b_4 X \times Y + b_5 Y^2 + e$, where Z was learning capability, X was human capital, and Y was social capital. Secondly, we ran polynomial regression in SPSS23.0, and the results of polynomial regression were presented in Table 4. Model3 showed that the slope of the line of capital congruence (HC = SC) as related to LC is given by 0.779(*p<0.01*), and curvature along the line of capital congruence as related to LC is assessed by 0.133(*p<0.01*). That is, *b1*

**Table 1. Means, standard deviation and correlation.**

| Variables | M | SD | 1 | 2 | 3 | 4 | 5 | 6 | 7 | 8 | 9 | 10 |
|---|---|---|---|---|---|---|---|---|---|---|---|---|
| *1. Age* | 2.84 | 1.129 | N | | | | | | | | | |
| *2. Stage* | 2.26 | 0.770 | 0.683** | N | | | | | | | | |
| *3. Type* | 3.59 | 1.958 | -0.110* | -0.080 | N | | | | | | | |
| *4. Size* | 3.20 | 1.463 | 0.786** | 0.617** | -0.192** | N | | | | | | |
| *5. HC* | 3.354 | 0.693 | -0.074 | -0.230** | -0.036 | 0.024 | (0.908) | | | | | |
| *6. SC* | 3.478 | 0.589 | -0.038 | -0.242** | -0.296** | 0.026 | 0.718** | (0.893) | | | | |
| *7. LC* | 3.472 | 0.520 | -0.186** | -0.251** | -0.158** | -0.136** | 0.602** | 0.704** | (0.908) | | | |
| *8. ED* | 3.396 | 0.694 | -0.108* | -0.206** | -0.216** | -0.111* | 0.203** | 0.386** | 0.233** | (0.806) | | |
| *9. EC* | 3.415 | 0.642 | 0.062 | -0.078 | -0.337** | 0.049 | 0.115* | 0.332** | 0.262** | 0.428** | (0.825) | |
| *10. IP* | 3.274 | 0.497 | -0.219** | -0.282** | 0.133** | -0.179** | 0.570** | 0.442** | 0.443** | -0.051 | -0.201** | (0.829) |

*P<0.05;

**P<0.01;

N = 456;

HC = Human Capital, SC = Social Capital, LC = Learning Capability, ED = Environmental Dynamism, EC = Environmental Competitiveness, IP = Innovation Performance;

Values in parentheses represent Cronbach's alpha coefficients.

+$b2$ indicated LC increases as HC and SC increase, and $b3+b4+b5$ suggested the agreement of HC and SC related to LC in a non-linear way. A significant negative $b1-b2$ indicated LC was higher when the inconsistency of HC and SC, such that SC was higher than HC, and vice versa. A significant positive $b3-b4+b5$ suggested LC would increase more sharply when the inconsistency of HC and SC was higher and higher.

Thirdly, we plotted the three-dimensional response surface to aid the interpretation of the results (in Fig 3). The graph indicated LC would be relatively high when HC and SC was inconsistency, such that LC was at highest level in low HC-high SC, or LC was at higher level in high HC-low SC than in low HC-low SC. Hypothesis2 was partly supported further.

In hypothesis5 we proposed that environmental dynamism and environmental competitiveness will jointly and positively moderates the relationship of capital congruence, learning

**Table 2. Hierarchical regression of main and mediation effects.**

| Variable | LC | | | IP | | | |
|---|---|---|---|---|---|---|---|
| | *M1* | *M2* | *M3* | *M4* | *M5* | *M6* | *M7* |
| *Control* | —— | —— | —— | —— | —— | —— | —— |
| *HC* | **0.593**\*** | | 0.194\*** | **0.563**\*** | | 0.472\*** | |
| *SC* | | **0.729**\*** | 0.566\*** | | **0.506**\*** | | 0.365\*** |
| *HC×SC* | | | **0.058**\* | | | | |
| *LC* | | | | | | **0.152**\*** | **0.193**\*** |
| $R^2$ | 0.414\*** | 0.526\*** | 0.547\*** | 0.379\*** | 0.301\*** | 0.393\*** | 0.319\*** |
| $R^2$ *adj* | 0.407\*** | 0.521\*** | 0.540\*** | 0.372\*** | 0.293\*** | 0.385\*** | 0.310\*** |
| *F* | 63.496\*** | 100.018\*** | 77.417\*** | 55.012\*** | 38.802\*** | 48.437\*** | 35.051\*** |
| *VIF* | 1.110 | 1.236 | 1.113 | 1.110 | 1.236 | 1.706 | 2.111 |

*P<0.1;

**P<0.05;

***P<0.01

**Table 3. Bootstrapping of mediation effects.**

| Outcome variable | Independent variable | Sobel test | Direct and indirect | Effects | se. | 95% *confidence interval* | |
|---|---|---|---|---|---|---|---|
| | | | | | | *LLCI* | *ULCI* |
| IP | HC | 3.201*** | *Indirect effect* | 0.067 | 0.020 | 0.029 | 0.106 |
| | | | *direct effect* | 0.341 | 0.034 | 0.274 | 0.408 |
| | SC | 4.381*** | *Indirect effect* | 0.154 | 0.030 | 0.095 | 0.216 |
| | | | *direct effect* | 0.218 | 0.049 | 0.122 | 0.314 |

capability and innovation performance. We firstly used variance inflation factors *(VIF) to* examine multiple-collinearity before testing hypothesis. The maximum of VIF was 1.228, below rule of thumb cut off of 10. Bootstrapping method (95% confidence interval, sample = 5000), displayed in Table 5, was utilized to test whether three-way interaction effect of learning capability, ED and EC was significant and positive. Four groups of participants were formed at the values of ED and EC, where high ED or high EC was one standard deviation above the mean, and low ED or low EC was one standard deviation below the mean. The results of bootstrapping showed in the group of high ED-high EC, capital congruence (*HC×SC*) and LC had significantly positive effect on IP (*β = 0.084, CI [0.034, 0.125]*). Similarly, we found the 95% bias-corrected confidence interval excluded zero in high ED-high EC and low ED-low EC groups, that is, the interaction effect of LC, ED and EC on IP was positive and significant. Hypothesis5 was supported.

Meanwhile, according to the method of Toothaker (1993), the moderating effect of ED and EC was graphed in Fig 4 to explore whether the interactions were in consistency with hypothesis. The simple slope showed the positive relationship between learning capability and innovation performance in four group of moderating combination, and the significance was strongest when ED was high and EC was high.

## Discussion

The current research has examined how human and social capital affect learning capability and innovation in R&D firms under dynamic and competitive environment, and the

**Table 4. The results of polynomial regression.**

| Variable | LC | | |
|---|---|---|---|
| | *M1* | *M2* | *M3* |
| *Control* | —— | —— | —— |
| $HC(b_1)$ | | 0.203*** | 0.119** |
| $SC(b_2)$ | | 0.576*** | 0.646*** |
| *HC Squared*$(b_3)$ | | | 0.001 |
| *HC×SC* $(b_4)$ | | | -0.205** |
| *SC Squared*$(b_5)$ | | | 0.337*** |
| $R^2$ | 0.096*** | 0.544*** | 0.588*** |
| $R^2$ *adj* | 0.088*** | 0.538*** | 0.580 |
| F | 12.014*** | 89.410*** | 70.870*** |
| VIF | 2.790 | 2.537 | 3.897 |
| b1+b2 | | | 0.779*** |
| b3+b4+b5 | | | 0.133*** |
| b1-b2 | | | -0.373*** |
| b3-b4+b5 | | | 0.543*** |

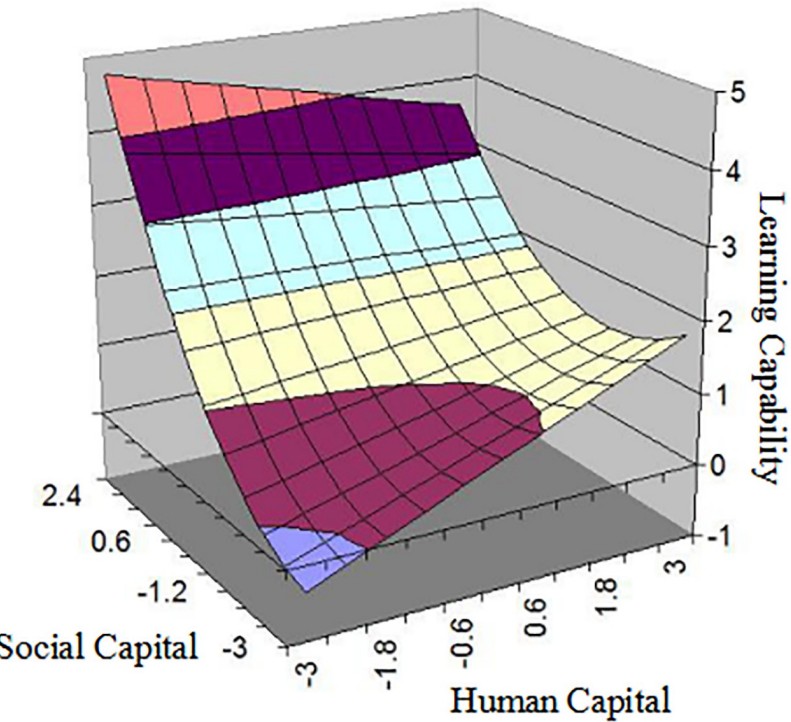

**Fig 3. The response surface of capital congruence and learning capability.**

synergistic effect between human and social capital (in this study, defined as capital congruence). Based on data from 456 R&D firms, we found that: (1) learning capability of R&D firms is higher when human capital is at low level and social capital is at high level (e.g. ② in Fig 2) than when they are at other situations (e.g. ① ③ ④ in Fig 2); (2) learning capability partially mediate the impact of human capital and social capital on innovation performance; (3) environmental dynamism and environmental competitiveness have a joint moderating effect on the relationship between learning capability and innovation performance. That is, the relationship is stronger when environmental dynamism and environmental competitiveness are at high levels.

## Theoretical contributions

This study has several theoretical contributions in the fields of organizational behavior and human resource management. Firstly, our findings reveal and clarify the effective relationship between human and social capital of R&D firms on learning capability, thus lending support

**Table 5. Bootstrapping of three-way interaction effects.**

| Outcome variable | Mediator variable | Independent variable | Mediated moderated effects | | | | |
|---|---|---|---|---|---|---|---|
| | | | Moderator variable (ED,EC) | Effects | se. | 95% confidence interval | |
| | | | | | | LLCI | ULCI |
| IP | LC | HC×SC | (High, High) | 0.084 | 0.024 | 0.034 | 0.125 |
| | | | (High, Low) | 0.083 | 0.025 | 0.031 | 0.127 |
| | | | (Low, High) | 0.022 | 0.014 | -0.002 | 0.057 |
| | | | (Low, Low) | 0.021 | 0.010 | 0.005 | 0.045 |

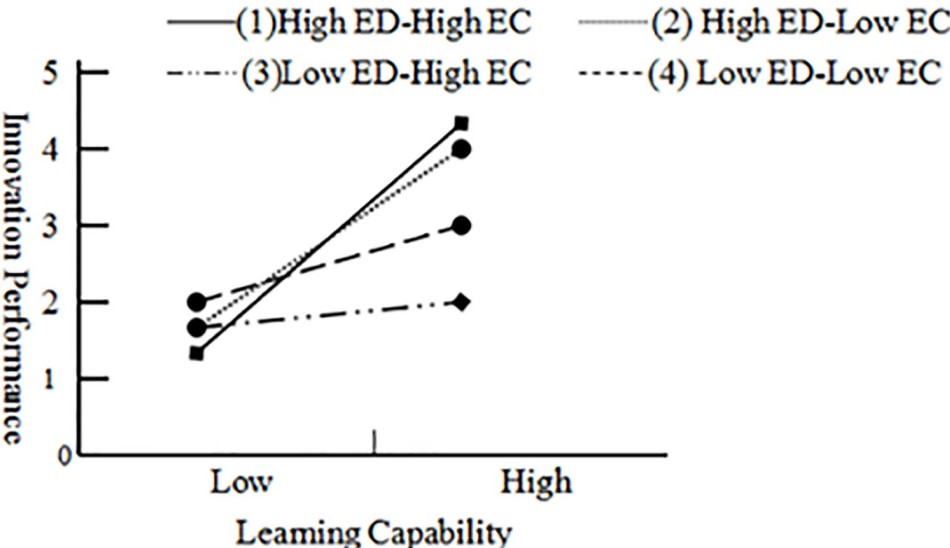

**Fig 4. Three-way interaction effect of environmental dynamism and competitiveness.**

for the competitive hypothesis between human capital and social capital [18, 19]. This is an interesting and counter-intuitive finding. Although the previous competitive hypothesis holds that the negative interaction between human capital and social capital can help enterprises to achieve better outcomes [19], we further find that not all the combinations of social capital and human capital can play an important role, while the combination of high social capital and low human capital can improve organizational learning ability. One possible explanation is that social capital may be substituted for human capital [19] to be an effective way to obtain organizational knowledge resources [23]. Social capital is clearly needed by R&D firms with low level of human capital to make most of learning and innovative capability [23]. Meanwhile, in contrast with expertise, skill and knowledge, this result underscores the support from network contacts is more crucial for strengthening organizational capability. However, human capital can not substituted for social capital to play the role of network.

Secondly, our findings illustrate how learning capability explains the relationship of human capital, social capital and organizational innovation performance. In R&D firms, innovation was regarded as one of the most important contributors to the organizational development and survival [24]. With human and social capital, organizations are more likely to strengthen learning capability in an innovative manner in order to absorb and deploy these resources effectively, which in turn, speed up the product renewal, process re-engineering and change of administrative mode. The finding indicates that organizations should do their best to utilize capital to promote organizational capability and to facilitate innovation.

Finally, we contribute to the recent development of learning capability and innovation, and expand the relationship to the complex context. Prior researches focus on the role of external environment (as environmental competitiveness) [58] or internal environment (environmental dynamism) [59] in the organizational management, and they are reflected simultaneously in few studies. Actually, in the practices of organizational management, organizations often face two types of environment with opposite tensions or paradoxes [50]. Our study suggests that environmental dynamism and competitiveness have a joint moderating effect on the relationship between learning capability and innovation performance, and this relationship may

be determined by various features of R&D firms, as innovational route fuzzy, innovational time pressing and innovational situation changing.

## Practical implication

There are several practical implications for strengthening organizational capability and improvement of innovation performance in R&D firms. Fist of all, according to our pre-researches from more than 500 Chinese R&D firms, we found the shortage of soft resources (e.g. talents and knowledge) has been the biggest obstacle for firms' sustainable development and survival. Thus, organizations should give priority to explore social capital (as information and financial resources) and then develop human capital (as knowledge and skill), which contribute to maximizing the organizational learning capability. Secondly, R&D firms must maintain and build competitive advantage through innovation, but knowledge absorption and application ability is the key driver for organizational innovation. Our finding indicates two aspects in regard to promote innovation. On one hand, organizations need to attach importance to resources involvement, especially social network resource. On the other hand, organizations should realize that knowledge and organizational capability is the driving force of innovation. Only by absorbing and acquiring new knowledge, integrating and upgrading existing knowledge, can organizations accomplish sustainable development. Thirdly, in response to uncertainty and changing environment, practitioners could take advantage of environmental factors. Once organizational innovation can not meet the market demand, enterprises can increase the environmental uncertainty and spread environmental dynamic and competitive information to create a threat awareness among employees. Generally, environmental dynamism and competitiveness can enhance the effect of learning capability on innovation in R&D firms.

## Limitations and future research

In this study, we contend that the relationship between human and social capital is competitive, and two types of capital have synergistic effect on organizational outcomes in the perspective of resource transformation. Meanwhile, we posit human and social capital can provide resource support for organizational innovation through learning capability under dynamic and competitive environment. However, there are several limitations that should be considered and remedied in future studies. Firstly, multiple-source survey was employed to collect data, and some statistical methods [60] have been utilized to verify common method variance, but the results may be biased as a result of subjectivity. Subsequent studies can take different raters into consideration to measure different variables, or collect data within multiple-time periods. Secondly, another limitation is that this study is conducted within a single high-tech industry. Thus, it remains unclear whether the conclusions can be generalized to other industries. We will do our best to take samples from other industries to generalize the results in the future studies. Finally, culture, as a core social and organizational background characteristic, may play an important role in organizational capital management. As Xiao and Tsui [61] described, culture on the organizational level has a potential constraining effects on social capital, especially in collectivistic society. Therefore, we should take the effect of culture into consideration in the future research.

## Conclusions

Nowadays, it is quite common for the uncertainty of the environment in the process of organizational development. And innovation is the key factor to navigate the uncertainty of the environment. Organizations can innovate and create only by relying on human capital (as

knowledge), social capital (as network) and learning capability of their staff. In previous studies, there are three contradicting theoretical claims on the relationship between human capital and social capital, that is, competitive, complementary, neither competitive nor complementary. In this study, the construct of capital congruence and in-congruence are proposed for the first time, which reflects the high or low level of synergistic effect between human and social capital. Our research indicates the relationship is competitive in R&D firms, and the findings not only reconcile the precious inclusive findings, but provide the first evidence of a significant relationship between human and social capital in complex environment. Meanwhile, we look forward to deepening the study related capital congruence, as its boundary condition, and fostering the researches on all kinds of capital in stimulating organizational outcomes in the future.

## Supporting information

**S1 Data.**
(SAV)

## Author Contributions

**Conceptualization:** Shuwen Li.

**Data curation:** Rui Sun.

**Formal analysis:** Shuwen Li.

**Funding acquisition:** Rui Sun.

**Methodology:** Shuwen Li.

**Software:** Shuwen Li.

**Validation:** Wei Liu.

**Writing – original draft:** Shuwen Li.

**Writing – review & editing:** Shuwen Li, Wei Liu.

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
