## [Decision Letter · Decision Letter 0]

30 Jan 2020

PONE-D-19-33171

A Congruence Perspective on How Human and Social Capital affect Learning Capability and Innovation

PLOS ONE

Dear Mr. LI,

Thank you for submitting your manuscript to PLOS ONE. After careful consideration, we feel that it has merit but does not fully meet PLOS ONE’s publication criteria as it currently stands. Therefore, we invite you to submit a revised version of the manuscript that addresses the points raised during the review process.

We would appreciate receiving your revised manuscript by Mar 15 2020 11:59PM. To enhance the reproducibility of your results, we recommend that if applicable you deposit your laboratory protocols in protocols.io, where a protocol can be assigned its own identifier (DOI) such that it can be cited independently in the future. For instructions see: http://journals.plos.org/plosone/s/submission-guidelines#loc-laboratory-protocols

We look forward to receiving your revised manuscript.

Kind regards,

Federica Angeli

Academic Editor

PLOS ONE

3. In your data availability statement you write, "All relevant data are within the paper and its Supporting Information files." Please ensure you have provided the individual data points used to create the figures and determine means, medians and variance measures presented in the results, tables and figures (http://journals.plos.org/plosone/s/data-availability#loc-faqs-for-data-policy). If these data cannot be publicly deposited or included in the supporting information, e.g. due to patient privacy or ownership by a third party, explain why and explain how researchers may access them.

4. Please ensure that you include a title page within your main document. We do appreciate that you have a title page document uploaded as a separate file, however, as per our author guidelines (http://journals.plos.org/plosone/s/submission-guidelines#loc-title-page) we do require this to be part of the manuscript file itself and not uploaded separately.

Reviewers' comments:

Reviewer's Responses to Questions

**Comments to the Author**

1. Is the manuscript technically sound, and do the data support the conclusions?

Reviewer #1: Yes

Reviewer #2: Yes

2. Has the statistical analysis been performed appropriately and rigorously? 

Reviewer #1: Yes

Reviewer #2: Yes

3. Have the authors made all data underlying the findings in their manuscript fully available?

Reviewer #1: Yes

Reviewer #2: Yes

4. Is the manuscript presented in an intelligible fashion and written in standard English?

Reviewer #1: Yes

Reviewer #2: Yes

5. Review Comments to the Author

Reviewer #1: The paper is well written and addresses an important literature gap on how human and social capital interacts with one another to influence learning and which in turn affects innovation performance. The analysis is also excellent. One issue I had was that the paper seems too straight forward. There are no counter-intuitive arguments or findings. I would urge the authors to think deeper to frame the questions in a more counter-intuitive manner. Also, I was hoping to look at the work of Xiao and Tsui work on how brokers do not work in China, given that it is a collectivistic society. But the paper does not refer to this critical piece of work. Xiao and Tsui have shown how social capital in China functions differently and thereby identifying boundary conditions for Burt's brokering form of social capital. From the work of Xiao and Tsui, one would assume that networks, and by extension, social capital would accumulate differently. If the authors, with some additional literature based on Chinese social capital and Chinese cultural settings, can identify counter-intuitive arguments or findings, the paper can be far more impactful.

Reviewer #2: 1. This paper discusses the topic about human capital, social capital affect learning capability have the significant importance and practicability.

2. I admire authors can use 456 company resources, it is a very useful data and hard to collect it.

3. But, the authors test the model fit and AVE show on Table 1, but I can’t find the validity examination (items coefficient). Please provide it.

4. Although this study used hierarchical regression analysis to test all of the hypotheses, but the authors didn’t use structural equation modeling (SEM). Therefore, I suggest the model fit and AVE test is not necessary. I suggest only use confirmatory factor analysis can be validity examination.

5. This paper to do moderate test, I want to know how to do control Variance inflation factor (VIF) test in hierarchical regression moderate test. If use mean center? please provide it

6. I want to know the reasons of hypothesis 2 was partly supported, please provide it.

7. In summary, this is a good study, the authors discusses the importance and practicability topic, and selected more good sample, I admire the researcher’s endeavor, but I also suggest the authors need to make up data analysis, and factor analysis need to be corrected, and improve the reasons of hypothesis 2 was partly supported will be fine.

6. PLOS authors have the option to publish the peer review history of their article (what does this mean?). If published, this will include your full peer review and any attached files.

Reviewer #1: No

Reviewer #2: Yes: Name - Hsiang-Heng Chen

Institution / Affiliation: Associate Professor at Sino-Australian International Hotel Management Collage, Tourism College of Zhejiang

Biography Summary: Dr. Hsiang-Heng Chen is currently Ph. D. in National Central University, Taiwan. He had technology manufacturing senior executive manager work experience and published more than hundred academic papers in the field of organization behavior, management, and personality. He research interest is in the field of entrepreneurship, and business administration.

E-mail: mikenrvs@qq.com

---

## [Author Response · Author response to Decision Letter 0]

11 Feb 2020

Referee:1

Review Comments: 

The paper is well written and addresses an important literature gap on how human and social capital interacts with one another to influence learning and which in turn affects innovation performance. The analysis is also excellent. One issue I had was that the paper seems too straight forward. There are no counter-intuitive arguments or findings. I would urge the authors to think deeper to frame the questions in a more counter-intuitive manner. Also, I was hoping to look at the work of Xiao and Tsui work on how brokers do not work in China, given that it is a collectivistic society. But the paper does not refer to this critical piece of work. Xiao and Tsui have shown how social capital in China functions differently and thereby identifying boundary conditions for Burt's brokering form of social capital. From the work of Xiao and Tsui, one would assume that networks, and by extension, social capital would accumulate differently. If the authors, with some additional literature based on Chinese social capital and Chinese cultural settings, can identify counter-intuitive arguments or findings, the paper can be far more impactful.

RESPONSE1: Thank you for your comment. Firstly, different from the structural hole in social capital discussed by Xiao and Tsui (2007), this study explores the relationship between human capital and social capital and how their combination affects organizational innovation, which determines our research finding about the relationship between human and social capital. In addition, We have to admit that culture is indeed an important factor affecting social capital, human capital and organizational innovation. As Xiao and Tsui (2007) described, culture on the organizational level has a potential constraining effects on social capital. However, we don’ t take culture into consideration in the research model. Therefore, the limitation is added to “Limitations and Future Research” in this study.

Secondly, an interesting and counter-intuitive finding was presented in this study. The competitive hypothesis suggests a negative interaction between human capital and social capital (Klyver and Schenkel 2013), while how to combine human capital with social capital can bring better organizational efficiency remains largely unexplored. Thus, a clear picture of the combination between human and social capital has yet to emerge. Our study shows that not all the combinations of social capital and human capital can play an important role in organizations, while the combination of high social capital and low human capital can better improve organizational learning ability. In other words, network resources are more valuable for organizations which suffer from shortfalls in human capital than for their counterparts with larger human capital, which challenges the previous researches noted human capital always benefits for organizations (Brymer et al., 2014). 

References

Xiao ZX, Tsui AS. When brokers may not work: the cultural contingency of social capital in Chinese high-tech firms. Administrative Science Quarterly, 2007; 52(1): 1-31. https://doi.org/10.2189/asqu.52.1.1.

Klyver K, Schenkel MT. From Resource Access to Use: Exploring the Impact of Resource Combinations on Nascent Entrepreneurship. Journal of Small Business Management, 2013; 51(4): 539–556. https://doi.org/10.1111/jsbm.12030.

Brymer RA, Molloy JC, Gilbert BA. Human capital pipelines: competitive implications of repeated interorganizational hiring. Journal of Management, 2014; 40(2): 483-508. https://doi.org/10.1177/0149206313516797.

Referee:2

Review Comments: 

Q1: This paper discusses the topic about human capital, social capital affect learning capability have the significant importance and practicability. I admire authors can use 456 company resources, it is a very useful data and hard to collect it. But, the authors test the model fit and AVE show on Table 1, but I can’t find the validity examination (items coefficient). Please provide it.

RESPONSE1: Thank you for your positive feedback. We have added internal consistency reliability (Cronbach’ s alpha coefficients) for these constructs in parentheses of table1, and factor loading is added to the “Measurements” part. 

Q2: Although this study used hierarchical regression analysis to test all of the hypotheses, but the authors didn’t use structural equation modeling (SEM). Therefore, I suggest the model fit and AVE test is not necessary. I suggest only use confirmatory factor analysis can be validity examination.

RESPONSE2: Thank you for your comment. We have deleted the AVE and model fit in table1.

Q3: This paper to do moderate test, I want to know how to do control Variance inflation factor (VIF) test in hierarchical regression moderate test. If use mean center? please provide it.

RESPONSE3: Thank you for your feedback. We use bootstrapping method to test the moderating effect of environmental dynamics and environmental competitiveness. And this method can not provide a procedure for collinearity testing. However, in view of the doubt of reviewer on the moderating effect, we used the hierarchical regression method to test the VIF. The results showed that the maximum of VIF was 1.228, below rule of thumb cut off of 10 (see page 17). Therefore, there is no serious collinearity in the moderating effect model.

Q4: I want to know the reasons of hypothesis 2 was partly supported, please provide it.

RESPONSE4: Thank you for your positive feedback. We add the reasons of hypothesis 2 was partly supported. As follows:

This is an interesting and counter-intuitive finding. Although the previous competitive hypothesis holds that the negative interaction between human capital and social capital can help enterprises to achieve better outcomes (Klyver and Schenkel, 2013), we further find that not all the combinations of social capital and human capital can play an important role, while the combination of high social capital and low human capital can improve organizational learning ability. One possible explanation is that social capital may be substituted for human capital (Klyver and Schenkel, 2013) to be an effective way to obtain organizational knowledge resources (Semrau and Hopp, 2016). Social capital is clearly needed by R&D firms with low level of human capital to make most of learning and innovative capability(Semrau and Hopp, 2016). Meanwhile, in contrast with expertise, skill and knowledge, this result underscores the support from network contacts is more crucial for strengthening organizational capability. However, human capital can not substituted for social capital to play the role of network.

References

Klyver K, Schenkel MT. From Resource Access to Use: Exploring the Impact of Resource Combinations on Nascent Entrepreneurship. Journal of Small Business Management, 2013; 51(4): 539–556. https://doi.org/10.1111/jsbm.12030.

Semrau T, Hopp C. Complementary or compensatory? A contingency perspective on how entrepreneurs’ human and social capital interact in shaping start-up progress. Small Business Economics, 2016; 46(3): 407-423. https://doi.org/10.1007/s11187-015-9691-8.

Q5: In summary, this is a good study, the authors discusses the importance and practicability topic, and selected more good sample, I admire the researcher’s endeavor, but I also suggest the authors need to make up data analysis, and factor analysis need to be corrected, and improve the reasons of hypothesis 2 was partly supported will be fine.

RESPONSE5: Thank you very much for your appreciation. We have revised our manuscript in response to your very helpful and constructive comments.

---

## [Decision Letter · Decision Letter 1]

25 Mar 2020

A Congruence Perspective on How Human and Social Capital affect Learning Capability and Innovation

PONE-D-19-33171R1

Dear Dr. LI,

We are pleased to inform you that your manuscript has been judged scientifically suitable for publication and will be formally accepted for publication once it complies with all outstanding technical requirements.

With kind regards,

Federica Angeli

Academic Editor

PLOS ONE

Additional Editor Comments (optional):

Reviewers' comments:

Reviewer's Responses to Questions

**Comments to the Author**

1. If the authors have adequately addressed your comments raised in a previous round of review and you feel that this manuscript is now acceptable for publication, you may indicate that here to bypass the “Comments to the Author” section, enter your conflict of interest statement in the “Confidential to Editor” section, and submit your "Accept" recommendation.

Reviewer #1: All comments have been addressed

Reviewer #2: All comments have been addressed

2. Is the manuscript technically sound, and do the data support the conclusions?

Reviewer #1: Yes

Reviewer #2: Yes

3. Has the statistical analysis been performed appropriately and rigorously? 

Reviewer #1: Yes

Reviewer #2: Yes

4. Have the authors made all data underlying the findings in their manuscript fully available?

Reviewer #1: Yes

Reviewer #2: Yes

5. Is the manuscript presented in an intelligible fashion and written in standard English?

Reviewer #1: No

Reviewer #2: Yes

6. Review Comments to the Author

Reviewer #1: I would have actually liked to see the culture argument in the main text than in the Limitation section. Could you please explore that? Other than that, I am ok with the paper now but it needs copy editing by an editor.

Reviewer #2: This is great research, the authors have adequately addressed my comments raised, I suggest "accept" this paper.

7. PLOS authors have the option to publish the peer review history of their article (what does this mean?). If published, this will include your full peer review and any attached files.

Reviewer #1: No

Reviewer #2: Yes: Name - Hsiang-Heng Chen

Institution / Affiliation: Associate Professor at Sino-Australian International Hotel Management Collage, Tourism College of Zhejiang

Biography Summary: Dr. Hsiang-Heng Chen is currently Ph. D. in National Central University, Taiwan. He had technology manufacturing senior executive manager work experience and published more than hundred academic papers in the field of organization behavior, management, and personality. He research interest is in the field of entrepreneurship, and business administration.

E-mail: mikenrvs@qq.com

---

## [Editor Report · Acceptance letter]

27 Mar 2020

PONE-D-19-33171R1 

A Congruence Perspective on How Human and Social Capital affect Learning Capability and Innovation 

Dear Dr. LI:

I am pleased to inform you that your manuscript has been deemed suitable for publication in PLOS ONE. Congratulations! Your manuscript is now with our production department. 

With kind regards,

on behalf of

Prof. Federica Angeli 

Academic Editor

PLOS ONE